



# A climatology of open and closed mesoscale cellular convection over the Southern Ocean derived from Himawari-8 observations

Francisco Lang[1], Luis Ackermann[1,2], Yi Huang[2,3], Son C.H. Truong[1,2], Steven T. Siems[1,2], and Michael J. Manton[1]

[1]School of Earth, Atmosphere and Environment, Monash University, Melbourne, VIC, Australia.
[2]Australian Research Council Centre of Excellence for Climate Extremes (CLEX), Melbourne, VIC, Australia.
[3]School of Earth Sciences, The University of Melbourne, Melbourne, VIC, Australia.

**Correspondence:** Francisco Lang (francisco.lang@monash.edu)

**Abstract.** Marine atmospheric boundary layer clouds cover vast areas of the Southern Ocean (SO), where they are commonly organized into mesoscale cellular convection (MCC). Using three years of Himawari-8 geostationary satellite observations, open and closed MCC structures are identified using a hybrid convolutional neural network. The results of the climatology show that open MCC clouds are roughly uniformly distributed over the SO storm track across mid-latitudes, while closed MCC clouds are most predominant in the southeast Indian Ocean with a second maximum along the storm track. The ocean polar front, derived from ECMWF-ERA5 sea surface temperature gradients, is found to be aligned with the southern boundaries for both MCC types. Along the storm track, both closed and open MCCs are commonly located in post-frontal, cold air masses. The hourly classification of closed MCC reveals a pronounced daily cycle, with a peak occurring late night/early morning. Seasonally, the diurnal cycle of closed MCC is most intense during the summer months (DJF). Conversely, almost no diurnal cycle is evident for open MCC.

## 1 Introduction

Marine atmospheric boundary layer (MABL) clouds play a primary role in defining the regional radiation budget over the Southern Ocean (SO) (Haynes et al., 2011) as they cover vast areas of the ocean surface (Trenberth and Fasullo, 2010) and exert strong shortwave and longwave radiative effects (Hartmann and Short, 1980). Despite the importance of MABL clouds, general circulation models (GCMs) and reanalysis products struggle to correctly simulate their complex microphysics and dynamics over the SO (Bodas-Salcedo et al., 2016; Kay et al., 2016). These biases commonly lead to the underestimation of shortwave radiation, in part because models produce less supercooled liquid water and lower cloud amount than observed, particularly in the cold sector of extra-tropical cyclones and in marine cold air outbreaks (Bodas-Salcedo et al., 2012, 2016; Field et al., 2014; Naud et al., 2014; Williams et al., 2013).



Satellite observations reveal that MABL clouds commonly exhibit different mesoscale morphology types, which are characterized by unique patterns of cloud organization. Based on the level of cellularity and mesoscale organization, Wood and Hartmann (2006) classified these clouds into open mesoscale cellular convection (MCC), closed MCC, no MCC, and cellular but disorganized clouds. More recently, Yuan et al. (2020) extended this classification by subdividing "no MCC" into stratus

clustered cumulus and suppressed cumulus, defining in total six types of organization. MCC morphology types are not only phenological classifications, but also an indication of underlying physical processes (Wang and Feingold, 2009b; Wood and Hartmann, 2006; Wood, 2012). These physical processes modulate fundamental features such as the overall cloud fraction and albedo, as well as microphysical properties such as precipitation rate, cloud droplet number concentrations and effective radius, affecting the radiation balance and precipitation efficiency of these clouds (Wood and Hartmann, 2006; Wood et al., 2011).

Ideal closed MCC clouds are stratocumulus clouds driven by longwave radiative cloud-top cooling and surface fluxes and are organized into distinctive patterns of hexagonally shaped cells with clear and descending edges. During summer months, shortwave heating at cloud-top has been observed to induce a diurnal cycle in stratocumulus clouds in the North Atlantic (e.g., Nicholls, 1984; Vial et al., 2019, 2021) and Pacific Oceans (Minnis and Harrison, 1984). The solar heating negates the longwave cooling at cloud-top and can thin the cloud deck, even to the point of cloud-break up (e.g., Lang et al., 2020; Nicholls,

1984; Minnis and Harrison, 1984). Overnight the moisture fluxes from the surface will help rebuild the cloud deck.

Open MCC are cumulus clouds arranged in hexagonal rings with a clear descending region in the center, particularly driven by surface forcing that creates and maintains this mesoscale morphology type (Atkinson and Zhang, 1996; McCoy et al., 2017; Wang and Feingold, 2009b). Open MCC clouds are commonly associated with a heavier drizzle, less short-wave reflectance and more transmissivity compared to closed MCC clouds (e.g., Ahn et al., 2017; Muhlbauer et al., 2014; Stevens et al., 2005;

Wang and Feingold, 2009b, a). Closed and open MCC types dominate the midlatitudes and subtropical stratocumulus decks (Muhlbauer et al., 2014), particularly across the Southern Ocean. In the midlatitudes, a transition is observed to occur from closed to open MCC, associated with the passage of extra-tropical cyclones and marine cold air outbreaks (Fletcher et al., 2016b; McCoy et al., 2017). The most studied mechanisms for this transition are cloud-aerosol-precipitation interactions and cold air advection over warmer water. The former can be thought of as microphysically driven, while the latter as large-scale

meteorologically driven (Yamaguchi and Feingold, 2015).

In-situ observations have revealed that open MCC cloud fields over the SO are commonly characterized by mixed-phase clouds (e.g., Lang et al., 2021), the frequent presence of drizzle/light precipitation (e.g., Ahn et al., 2017) and active secondary ice production (e.g., Huang et al., 2017). Lang et al. (2021) used shipborne observations to further demonstrate that at near-surface level, precipitation from open MCC commonly is associated with reduced temperatures or "cold pools," which are

driven by the evaporation of precipitation in the subcloud layer. Over the SO, closed MCC have been linked to non-drizzle conditions and aircraft observations showed that they are commonly found when a high-pressure ridge is the dominant meteorological feature (Ahn et al., 2017).

To analyze morphology types and associated cloud properties of open and closed MCC, numerous previous studies have developed cloud classification algorithms, commonly employing artificial neural networks (ANN). Wood and Hartmann (2006)

trained a three-layer neural network on the power spectra and probability density functions (PDF) of liquid water path. The





ANN analyzed subscenes of retrievals from the National Aeronautics and Space Administration (NASA) Moderate Resolution Imaging Spectroradiometer (MODIS) Aqua satellite to determine cloud morphologies (Platnick et al., 2003). However, their data were limited to only warm clouds for two months and did not include the SO. Muhlbauer et al. (2014) and McCoy et al. (2017) applied the same ANN classification method to a much more extensive data set (global scale for 1 year) to analyze

morphology types and associated cloud properties. McCoy et al. (2017) specifically explored relationships between air-sea temperature difference, estimated inversion strength (EIS), and marine cold air outbreaks for open and closed MCC clouds. They found a strong correlation between the marine cold air outbreaks and occurrence of both open and closed MCC in the midlatitudes. More recently, Watson-Parris et al. (2021) employed a convolutional neural network (CNN) to detect open MCC clouds from MODIS Terra observations and estimate their radiative impact. Rampal and Davies (2020) also employed a CNN

using Multiangle Imaging SpectroRadiometer (MISR) satellite observations (Diner et al., 1999) to investigate the relationships between MCC types and MABL cloud albedo over the Pacific, Indian, and SO regions. In particular, they established a relationship between cloud albedo and cloud heterogeneity as a direct function of the MCC type. Further, they found significantly lower frequency of occurrence of closed MCC (below 5%) at high latitudes compared to McCoy et al. (2017) and Muhlbauer et al. (2014). Their domain, however, did not include the portion of the SO between Australia and the Antarctica.

The main objective of this study is to develop a new classification algorithm employing a CNN to determine the climatological distribution of open and closed MCC clouds over the SO. We use Himawari-8 high-frequency geostationary satellite observations to examine the characteristics of open and closed MCC clouds within the context of the synoptic meteorology, specifically in relation to extra-tropical cyclone and cold fronts. The most important advantage of using Himawari-8 images is the high temporal resolution compared to MODIS and MISR. This temporal resolution allows us to have the ability to look at

the diurnal cycle of the MCC clouds over the SO, which has never been undertaken over this region for any type of MABL. The focus is to understand the mesoscale organization under post-cold frontal conditions and mechanisms that might explain the distribution and seasonality of these MCC cloud types, given that the largest model bias has been linked to this sector.

## 2    Data and methodology

### 2.1    Data source and domain

The observational data for this study are from the Advanced Himawari Imager (AHI) on board the Himawari-8 geostationary meteorological satellite (Bessho et al., 2016). Launched by the Japanese Meteorological Agency and becoming operational in July 2015, this satellite covers the Asia-Oceania region including a large portion of the SO. Himawari-8 products are available on the Japan Aerospace Exploration Agency (JAXA) P-Tree system. Himawari-8 provides a spatial resolution of 1-5 km and temporal resolution of 10 min. Reflectance from channels 1 (0.47 $\mu$m), 2 (0.51 $\mu$m), and 3 (0.64 $\mu$m), brightness temperature

from channel 10 (7.3 $\mu$m) and cloud effective radius, cloud optical thickness, and cloud-top height from the Himawari-8 cloud product are used as control, filtering, and contextual information for building up the manually labeled training data set. Different infrared channels were tested as inputs to the neural network, with Channel 11 (8.6 $\mu$m) having the best performance. Only 5 km resolution brightness temperature from channel 11 in an orthogonal gridded projection was used for model training





and subsequent MCC climatology classification. The domain selected for the study is between 80°E and 160°W and between

20° and 60°S, which covers portions of the Pacific, Indian, and SO regions, as illustrated in Fig. 1. This domain encompasses the area of the SO storm tracks in the midlatitude that directly affects Australia and New Zealand's weather, it is part of the largest international multi-agency effort called the Southern Ocean Clouds, Radiation, Aerosol Transport Experimental Study (SOCRATES, McFarquhar et al., 2021), and is characterized by a high density of extra-tropical cyclones and cold fronts (e.g., Hoskins and Hodges, 2005; Simmonds and Keay, 2000).

## 95   2.2   MCC cloud classification

Following McCoy et al. (2017), we focus on exploring the influence of the synoptic meteorology on open and closed MCC morphologies over the SO. The first step is to develop an algorithm to classify MCC clouds over the SO. As mentioned above, Wood and Hartmann (2006) first implemented an ANN for MCC morphology identification. More recently, several studies have applied a more advance neural network model based on convolution tensor operations in a convolution neural network

(CNN) for the identification and classification of MCC clouds (e.g., Rampal and Davies, 2020; Watson-Parris et al., 2021; Yuan et al., 2020). In deep learning, CNN models have been able to separate complex patterns into different categories. As Rampal and Davies (2020) point out, a deep-learning method based on spatial patterns is likely more advantageous because it can use a direct satellite channel for model training rather than an inferred product such as liquid water path (Wood and Hartmann, 2006).

## 105   2.2.1   CNN model structure

Our classification scheme of MABL clouds is based on a hybrid CNN model, which uses observations from a geostationary satellite to classify the observed domain as open MCC, closed MCC, or other. The category "other" is used for other cloud types, ocean and land. In contrast to other studies, we did not perform a separation into more cloud categories due to the limited capacity of hourly data processing. Specifically, the inputs to the CNN model consist of hourly data from 2016 to

2018 of brightness temperature (Channel 11) from AHI Himawari-8 at 5 km resolution. The hybrid nature of the model comes from having both scalar and spatial input layers, where the spatial input is a window of the brightness temperature, while the scalar inputs are the solar and satellite zenith and azimuth angles. The window of brightness temperature is meant to provide enough morphological information about the cloud configuration, while the angles provide the model with information regarding distortions in the viewing and irradiation angle. Adding this information to the model showed improved accuracy.

From the AHI domain, each grid point is classified by providing the hybrid model with a 2D array of the normalized brightness temperature (Channel 11) centered at the point to be classified and the corresponding satellite and solar angles for that grid point. After a sensitivity analysis, a window of 16 × 16 points (~80 km × 80 km) was used for the brightness temperature array, which provided the highest accuracy with the lowest computational cost. In effect, each grid point is classified using information of that grid point, the 255 surrounding points, the satellite viewing angle, and the solar angle. The model structure

is composed of three convolutional layers that process the spatial input (brightness temperature array), two layers that process the angles, and two layers applied after the output from the convolutional layers and the angle layers are concatenated. The





output of the model is a three-element vector whose elements sum up to 1, and are interpreted as the probability of the tested point to correspond to one of the three classes. The point is assigned the category corresponding to the element with the highest probability.

### 2.2.2 Training dataset

The model was trained in a supervised fashion, using a data set created by manually identifying areas where only open MCC, closed MCC, or neither were exclusively present, in a similar methodological manner used in previous studies (Rampal and Davies, 2020; Watson-Parris et al., 2021; Wood and Hartmann, 2006; Yuan et al., 2020). In order to ensure that the labeling of the open and closed MCC was consistent, the structure of the MCC clouds must follow the conservative criterion that open MCC must be open cell cloud, which looks "stringy", while the closed MCC must be closed cell cloud, which looks "bubbly" (Watson-Parris et al., 2021). The transitions from open to closed MCC clouds were, by default, classified as other. These areas were selected from approximately 400 independent scenes between January 2016 and December 2018, being carefully chosen such that relatively equal number of samples are taken from all seasons, allowing for a wide range of synoptic meteorology, solar zenith angles, and diurnal variation. All MCC areas selected are predominantly low-level clouds, defined as cloud-top height less than 3.5 km; while the areas representing the others class were selected from all other scenes, including areas with no clouds and land. These labeled areas accounted for  2.7 million individual pixels; ∼1.2 million under the open MCC category, ∼0.6 million under the closed MCC category, and the remaining (∼0.9 million) under the others category.

### 2.2.3 Training performance

In terms of accuracy, the model's training reaches a plateau fairly quickly, within about 45 iterations through the whole dataset (epochs), with a maximum training and validation accuracies around 93.7% and 4.1%, respectively (Fig. 2). The confusion matrix of the validation data set is shown in Table 1. This matrix displays a summary of the prediction results. The trained model shows an average precision of about 89% across the different types, with open MCC category exhibiting the lowest accuracy mainly due to having the lowest training sample size.

### 2.3 The SO meteorology and polar front

The SO meteorology is strongly influenced by the storm track, which is characterized by frequent and deep midlatitude cyclones that drive persistently strong zonal winds (Mace et al., 2009; Mace and Zhang, 2014). MABL clouds are commonly present in the cold sector of extra-tropical cyclones and in marine cold air outbreaks (Field et al., 2011; Kay et al., 2016; Naud et al., 2014; Williams et al., 2013). Figure 3a shows the frequency of winds exceeding 20 m s$^{-1}$ from the European Center for Medium-Range Weather Forecasts (ECMWF) ERA5 reanalysis across our domain (Hersbach et al., 2020), where midlatitudes and high latitudes are characterized by frequent high wind speeds. The cold sector located northwest of the cyclone center is a region of large-scale subsidence dominated by MABL clouds, where the inversion strength and cloud fraction are related because the inversion controls the mixing at cloud top (Klein and Hartmann, 1993). Over the post-cold-frontal SO regions, a strong





inversion has been observed (more stable conditions), which is favorable for the generation of shallow convection (Lang et al., 2018). About 80% of the marine cold air outbreaks occur in association with the passage of cyclones and they are characterized by a large air-sea temperature difference, where the cold air masses impinge on warmer midlatitude air (Papritz et al., 2015). The warm water-cool air contrast increases the flux of energy and moisture from the surface into the boundary layer, which influences development of MABL clouds (Abel et al., 2017; Fletcher et al., 2016a). The strength of the SO turbulent heat flux is strongly controlled by marine cold air outbreaks (Papritz et al., 2015). In the high latitudes and midlatitudes, a transition between closed MCC clouds to open MCC clouds occurs with the passage of cyclones and cold air outbreaks (McCoy et al., 2017).

Inatsu and Hoskins (2004) used global circulation models to demonstrate that the major determinant of the lower-troposphere storm track intensity over the SO was the enhanced midlatitude sea surface temperature (SST) gradients, or polar front (Dong et al., 2006; Moore et al., 1999). Dong et al. (2006) defined the polar front as the strong SST gradient, where a strong gradient is determined to be the southernmost location at which the SST gradient exceeds $1.5 \times 10^{-2}$ °C km$^{-1}$. Figures 3b and c show the mean SST and the SST meridional gradient from ERA5 reanalysis products between 2016 and 2018. The maximum SST gradient varies spatially in its mean position (Fig. 3c). The mean SST gradient path is further north in the Indian Ocean sector at ~43°S and moves poleward until it reaches ~57° S at 150°E. This north-south range of the mean SST gradient path is about 15°. Although the definition of polar front in Dong et al. (2006) differs from our estimates in Fig 3b, the mean polar path is consistent with the SST gradients from ERA5 and within the variability that corresponds to different observations and reanalysis products as shown in Dong et al. (2006).

### 2.3.1 Synoptic data

The relationships between MCC clouds and two synoptic features common to the SO storm track, namely cold fronts and extra-tropical cyclones, were explicitly studied. These features were calculated using ERA5. Extra-tropical cyclones were identified using the cyclone detecting and tracking algorithm developed by Pezza et al. (2008) and Murray and Simmonds (1991). This identification is based on the 3-hourly mean sea level pressure (MSLP). The algorithm transforms the MSLP latitude-longitude grid to a polar stereographic grid and then searches for the local maximum in the Laplacian of the MSLP field. Each cyclone identified is assigned as "open" or "closed" based on whether it has an open or closed isobar around the minimum. To select only meteorologically significant systems the pressure minimum had to satisfy a strength criterion: those between 0.2 and 0.7 hPa (°lat)$^{-2}$ were classified as "weak," and those with the strength greater than 0.7 hPa (°lat)$^{-2}$ were classified as "strong"; see Lim and Simmonds (2007) for complete details. Here, we use the term cyclone to refer to a specific feature at a specific time, rather than a complete life cycle. Over our domain and for the study period a total of 22,690 strong cyclone centers were identified.

The objective identification of cold fronts is based on the method developed by Hewson (1998) and improved by Berry et al. (2011). This algorithm identifies frontal points along the maximum of the horizontal gradient of the wet-bulb potential temperature at 850 hPa. The diagnosed fronts are then categorized into cold, warm, and quasi-stationary fronts according to





different speed ranges. The analysis by Berry et al. (2011) with ECMWF ERA-Interim reanalysis found the highest front frequency in the midlatitude storm tracks over the SO.

### 2.3.2  MCC cloud composites

For each cyclone center identified, we extracted the MCC classification for each grid point in a 3000 3000 km square centered
190    on the cyclone core to construct the composite structure. This cyclone-center composite allowed us to define a frame of reference where the cold-air side of the cyclone is commonly located in the northwest and southwest quadrants (e.g., Bodas-Salcedo et al., 2014; Lang et al., 2018; Truong et al., 2020).

The distance from a given grid point to the nearest cold front is defined as the distance along a line between the two that is aligned along the wind vector at the grid point. Considering that MCC clouds are mostly located in the cold sector, cold
195    fronts have to be eastward of the MCC systems within a distance of °, where a MCC system is defined as a continuous group of grid points classified as either closed or open MCCs. For the wind direction, we use ERA5 wind components at 850 hPa. The frequency of open and closed MCC cloud is estimated by distance into 100 km bins, to produce a composite across the cold sector. During the period from 2016 to 2018 a total of 25,654 open MCC systems and 15,722 closed MCC systems were associated with a cold front.

200    ### 2.4  Examples of the classification

Two examples of classified brightness temperature images under a post-frontal environment for winter and summer seasons in the midlatitude are shown in Fig. 4. The summertime scene (Fig. 4a) shows a cloud field of MCC clouds in the cold sector of an extra-tropical cyclone located at ∼59°S. The cloud field shown in this example is midway through a transition from closed to open MCC cloud, where closed MCC are moving from high latitudes advected over a warmer ocean. Similarly,
205    Fig. 4c shows an example for wintertime, where a large high-pressure system is present over Australia according to the mean sea level pressure from the Australian Bureau of Meteorology (not shown). Located in the south edges, the MCC cloud field displays a transition from closed to open MCC cloud followed by frontal clouds. The classification results are overlaid on two subscenes of the Channel 11 brightness temperature image (Fig. 4b, d), where low-cloud-dominated areas show the presence of the two morphology types. The areas not classified correspond to others; for example, Fig. 4b shows a group of clouds south
210    and southeast that are mostly altostratus clouds. For these two examples, one can visually confirm that the CNN performs reasonably well in selecting the open and closed MCC morphologies and their transitions.

## 3  Results

The CNN model was run on all the hourly brightness temperature images over the domain in Fig. 1 between 2016 and 2018. For this period, 25,494 images were processed and classified into the categories open MCC, closed MCC and others.





## 3.1 MCC climatology

The geographical distribution of the annual relative frequency of occurrence of open and closed MCC clouds is illustrated in
Figures 5a and b, respectively. The frequency of occurrence of MCC clouds is defined as the number of times a cloud type
(e.g., open MCC cloud) is observed in a grid point and time period divided by the total time. First, it is noted that the spatial
distribution of MCC clouds features a ~15° broad band across the domain. This band is located further south compared with
the Southeast Indian Ocean perhaps due to the influence of the Australian and New Zealand land masses. This is consistent
with the distribution of low-level clouds from CloudSat/CALIPSO data in Muhlbauer et al. (2014), which showed low-cloud
fraction peaks south of Australia and lower frequencies towards high latitudes.

Figure 5a shows that open MCC clouds exhibit a relatively uniform distribution across midlatitudes. They peak in the area
of the storm track between 40° and 50°S and have two local maxima over the surrounding ocean west of Tasmania (23%) and
the Tasman sea (25%). The presence of open MCC over the storm track is likely associated with marine cold air outbreaks and
frontal passages. While the closed MCC clouds are less frequent than the open MCC, they are most predominant (12%) over the
Southeast Indian Ocean (Fig. 5b) where persistent stratocumulus decks have been observed by previous studies (e.g., Atkinson
and Zhang, 1996; Klein and Hartmann, 1993; Muhlbauer et al., 2014). This region is located in the large-scale subsidence region
west of Australia, commonly influenced by strong high-pressure systems and upwelling of cold oceanic waters (Atkinson and
Zhang, 1996). As with open MCCs, closed MCC clouds are likely associated with marine cold air outbreaks in these regions.
Overall, the contributions of closed MCC are considerably lower, with frequency of occurrence ranging from about 5 to 12%.
A shift from closed to open MCCs clouds is seen from the Southeast Indian Ocean into the SO immediately south of Australia,
likely indicating that the stratocumulus clouds moving from the west break up into shallow cumulus clouds. Further poleward,
the occurrence of both MCC types tends to decrease with a slightly higher presence of closed MCC.

A blocking effect of New Zealand is observed eastward of ~170°E, as shown by a considerable decrease of the frequencies
for both MCC types. Similarly, the area eastward of Tasmania presents lower frequencies due to a land effect from the island. A
strong relationship between the MCC classifications and the SST gradients over the SO is seen in Fig. 3c and 5. The location of
the south boundaries for both classifications clearly shows an alignment with the maximum SST gradients over the domain. We
also notice that low frequencies for closed MCC are associated with low SST gradients; for instance, a band between 40°-50°S
and 100°-140°E shows a local minimum for both closed MCC occurrences and SST gradients. This relationship emphasizes
the temperature contrast between the cold air moving from high latitudes above relatively warmer water, creating a dynamically
favorable condition for MABL cloud development. At high latitudes, the occurrence of both morphologies tends to decrease
with a slightly higher presence of closed MCC. We note at the high latitudes, poleward of the polar ocean front, mid-level
clouds are commonly present (Mace et al., 2009; Truong et al., 2020), which obscures the observation of any boundary layer
clouds from passive satellite instruments. This reduction in the MCC frequencies may also be related to the higher wind speeds
at surface level as shown in Fig 3a. Highest wind speed frequencies of winds exceeding 20 m s$^{-1}$ are at the western portion
of our domain from ~80° to 100°E and southward of New Zealand from ~160°E to 170°W. These two regions correlate well
with a reduced fractional MCC cloud cover (Fig. 5). A local maximum is observed northeast of New Zealand; this region





coincides with a local maximum of cold fronts associated with the the South Pacific Convergence Zone (Berry et al., 2011).

However, MCC classification by Rampal and Davies (2020) shows primarily the occurrence of disorganized MCC in this area. We believe that our model is struggling to separate open from disorganized MCC over this area. Uncertainties in the separation of disorganized and open MCC using a CNN was also reported by Yuan et al. (2020).

The seasonal cycle of frequency of occurrence for MCC classifications is shown in Fig. 6. A considerable seasonal cycle is found for open MCC. The maximum frequency of occurrence of open MCC is found during the spring season (SON) over the

Tasman sea between 35° and 40°S (28%). Similarly, west of Tasmania and South Pacific Ocean between about 30° and 40°S, open MCC have higher frequencies above 25% of the time. During summer (JJA), open MCC frequencies are lower, with a considerable reduction in the frequency of occurrence in sectors such as the South Pacific Ocean and the Southeast Indian Ocean (∼15% during summer). A shift of the maximum further poleward is also observed during the summer season likely due to the influence of the Hadley cell extending further poleward, as does the storm track. The strong seasonality in the open

MCC frequency might be linked to the lower frequency of occurrence of cold-air outbreaks and the associated advection of cold air over warmer ocean surfaces, both reaching the minimum during summer months.

Compared to open MCC, the occurrence frequency of closed MCC shows less interseasonal variability. Closed MCC maxima are present over the Southeast Indian Ocean with a peak of 13% during summer. For west of Tasmania, Tasman sea and South Pacific Ocean regions, summer shows a narrower band of closed MCC frequencies compared to the other seasons. Similarly,

to open MCC clouds, the frequency peak moves poleward during summer along with the storm track.

### 3.1.1   Diurnal cycle

Figure 7 shows the diurnal cycle of the frequency of occurrence for annual mean and sorted by season for a latitudinal band between 40° and 50°S. Looking at the annual mean (Fig. 7a), the diurnal cycle of closed MCC exhibits a pronounced daily cycle with distinct 24 hr phasing. A maximum is found during night and/or early morning with a peak of 14% before sunrise

at 0400 local standard time (LST), a minimum at 1400 LST (9.9 %). At approximately sunset, the mean observed occurrence reaches its lowest point below 10% and increases through the night until approximately sunrise, with a range of the cycle of ∼4%. The standard deviation shows that the variability is approximately constant around 5% throughout the day. While a diurnal cycle was identifiable in all seasons for closed MCC, it was most intense during the warmer months, the Austral summer (DJF) and spring (SON), as would be expected. For the winter (JJA) and autumn (MAM) seasons, the diurnal cycle is

relatively flat through much of the day. The standard deviation also shows a low and constant variability approximately around 5% throughout the day for all the seasons.

In contrast, the diurnal cycle of open MCC is less distinct with a maximum of 25% in the afternoon at 18 LST. Compared to the closed MCC, the standard deviation for open MCC shows more variability throughout the day (∼10%). The open MCC occurrence shows higher afternoon peaks for the Austral winter and spring with frequency of occurrence higher than 30%

throughout the day, while summer and autumn are relatively flat through much of the day and at frequencies lower than 20%. The seasonal standard deviation shows larger differences between summer and the other seasons, with a low variability during the summer season (∼2%) and much higher for winter, spring and autumn (∼10%).





## 3.2 MCC relationship to extra-tropical cyclones and cold fronts

In this section, we investigate the main characteristics of the MCC clouds relative to the extra-tropical cyclones and cold fronts.
The role of both cyclones and cold fronts is analyzed to find a relationship between these synoptic conditions and MCC clouds that can explain the annual variability in the spatial pattern frequency and MCC cloudiness.

First, we look at the relationship between extra-tropical cyclones and MCC clouds using cyclone-center composites. The frequency of open MCC (Fig. 8a) has a maximum equatorward of a low pressure center and westward of the cold frontal zone, and lower frequencies poleward. This maximum reaches 22% about ~900 to 1300 km from the cyclone center. This sector on
the western side of the cyclones is on average a region of colder temperatures, lower moisture amounts, and lower precipitation than east of the low (e.g., Bauer and Del Genio, 2006; Lang et al., 2018; Naud et al., 2014; Truong et al., 2020). A lower frequency of occurrence for open MCC is observed across the warm frontal zone, and open MCC extends into the warm sector, on the eastward side of the low pressure center, with frequencies between 2% and 10%. We examine the seasonal cycle to help determine the synoptic factors in open MCC cloud development (Fig. 9). The peak concentration of open MCC is found to
be 27% during the winter season with the peak being at ~1200 km from the cyclone core. The lowest frequencies are found during the summer season with a peak of 19%. The distance of the peaks from the cyclone centre shows a small seasonal shift, moving further away of the 1000 km during the winter and spring seasons to closer distance between 800 and 1000 km during the summer and autumn seasons. This shift in the location of the peak is likely to be related to the larger extension and intensity of the wintertime extra-tropical cyclones (Simmonds and Keay, 2000), with more open MCC generated further away from the
low centers.

The closed MCC cloud maximum tends to occur on the western side of the low centers and in the wrap-around sector at the southwest of the open MCC, but with a much lower frequencies that peak at 7%. The peak frequency is also located farther away from the cyclone core (Fig. 8b) than that of the open MCC (at 1300 km). Winds over this area are primarily cold air from southwest, indicating that closed MCC clouds move behind the open MCC clouds. According to Naud et al. (2014), the
west side of the low center is characterized by low-level clouds, where the average cloud-top height using Multiangle Imaging Spectroradiometer (MISR) observations is found to be below 3 km. The frequency of closed MCC type shows a much weaker inter-seasonal variability (Fig. 9) as compared to that of the open MCC. Slightly higher frequencies are found during the winter season with a peak of 8.1%, while the minimum peak is observed during the summer season (6.7%). Note that it is likely that the occurrence of closed MCC is higher outside of the 1500 × 1500 km window of the figures. The timing and location of
the open MCC cloud seasonality around the cyclone centers is consistent with the connection to marine cold air outbreaks. Over the high latitudes and midlatitudes, the marine cold air outbreak frequencies peak in hemisphere winters (Fletcher et al., 2016b).

To explore the distribution of the MCC morphologies under a post-cold-frontal environment, we focus our analysis in the cold sector using as reference the distance from cold fronts. We only consider MCC systems that are located west of the cold
front; in total, 25,654 open MCC systems and 15,722 closed MCC systems were associated with a cold front. Figure 10 shows the frequency of occurrence sorted by distance into 100 km bins for open and closed MCC. The histogram for open MCC





shows the highest frequencies within 300 to 600 km from the cold front line, reaching a maximum between 400 and 500 km (7.7%). Beyond the 700 km distance, the frequency of open MCC decreases with distance from the cold front. The maximum for the closed MCC histogram is located approximately between 500 and 800 km distance, with a peak of 8.5% between 700
and 800 km. The results show a clear difference in the location of the maximum for each MCC type. This difference in the location of the maximum is consistent with cyclone-center composites and the examples in Fig. 2, where open MCC clouds are moving ahead of the closed MCC along with the mean flow, consistent with McCoy et al. (2017). For both morphologies, the histograms show low frequencies immediately behind the frontal line. A clear band behind cold fronts was first observed during the Aerosol Characterization Experiment 1 (ACE-1) campaign in the 1990s (Bates et al., 1998; Suhre et al., 1998). More
recently, Lang et al. (2021) described the same clear band during the Clouds Aerosols Precipitation Radiation and atmospheric Composition over the Southern Ocean (CAPRICORN) I research voyage. Significant changes in the distribution of open and closed MCC between the seasons are not observed (not shown).

## 4    Discussion and conclusions

High-frequency geostationary satellite observations over the Southern Ocean (SO) are used to explore how marine atmospheric
boundary layer (MABL) clouds are organized in mesoscale cellular convection (MCC) morphologies. We first focus on developing a convolution neural network (CNN) model to identify and classify open and closed MCC clouds based on three years of Himawari-8 satellite data from 2016 to 2018, and then to study their relationship to synoptic systems over the SO. The climatology showed that open MCC clouds are roughly uniformly distributed over the storm track across midlatitudes and have local maxima over the surrounding ocean west of Tasmania and New Zealand, 23% and 25% of the time respectively. While
closed MCC clouds are most predominant in the southeast Indian Ocean (12% of the time), an area characterized by persistent stratocumulus decks (Klein and Hartmann, 1993). Our results find that closed MCC clouds are less prevalent at high latitudes than found in previous studies using MODIS (McCoy et al., 2017; Muhlbauer et al., 2014). The algorithm used in Muhlbauer et al. (2014) and McCoy et al. (2017), however, is limited in that it only used liquid water path retrievals to classify the MCC cloud type (Wood and Hartmann, 2006). Over mid and high latitude oceans, the common presence of ice particles in clouds
(e.g., Huang et al., 2017) and precipitation poses a significant challenge to this method. More recent studies have used CNN models to also show a weak presence of closed MCC at high latitudes (e.g., Mohrmann et al., 2021; Rampal and Davies, 2020). For example, Rampal and Davies (2020) found that closed MCC has considerably lower frequency of occurrence (below 5%) at high latitudes, while stratus clouds are the dominant MABL cloud type with frequencies of occurrence ranging from about 20 to 35%. These differences from McCoy et al. (2017) and Muhlbauer et al. (2014) can be attributed to a number of factors, such
as differences in instrumentation, spatial resolution and sampling periods. Nonetheless, our classification had the advantages that we used a more advanced neural network technique with samples located exclusively over our domain and from the fields of brightness temperature, which are less prone to retrieval errors at high solar zenith angles.

The climatological frequency of occurrence of open and closed MCC clouds showed a strong relationship to the sea surface temperature (SST) gradients. In regions of enhanced surface forcing due to the warmer ocean-colder air temperature contrast,





the SST has been established as a driver mechanism for open MCC cloud development (McCoy et al., 2017). The maximum gradients of SST from ERA5 are aligned to the location of south boundaries longitudinally for both MCC types. When cold air from the Antarctica moves equatorward over the polar front (i.e. marine cold air outbreaks), the strong SST gradient increases the flux of energy and moisture from the surface into the boundary layer and facilitates the development of MABL clouds (Abel et al., 2017; Brümmer, 1996; Fletcher et al., 2016b). McCoy et al. (2017) point out that these stronger fluxes denote a transition

between closed MCC clouds from high latitudes to open MCC clouds. However, as mentioned above, our results showed lower frequencies at high latitudes, consistent with Rampal and Davies (2020) and Yuan et al. (2020). The lower frequencies of MCC morphologies at high latitudes are also consistent with the common presence of multilayer cloud structures within and above the MABL (Mace et al., 2009; Truong et al., 2020), which represents a challenge for the identification of MCC clouds using only cloud top observations. In addition, we noted that MCC cloud cover in this region might be influenced by the frequent

high wind speeds. The frequency of winds exceeding 25 m s-1 using ERA5 showed highest wind speed frequencies are in the western portion of our domain from ∼80° to 100°E and southward of New Zealand from ∼160°E to 170°W. These two regions correlate well with a reduced fractional MCC cloud cover (Fig. 5). An open question related to this is whether frequent and strong winds disrupt the formation (or maintenance) of ideal open and closed MCC clouds, which is worthy of future research and explanation.

The hourly classification from AHI Himawari-8 Channel 11 brightness temperature allowed the study of the diurnal cycle of both MCC morphologies (Fig. 7). The diurnal cycle of MABL clouds has been documented for decades, where a strong diurnal cycle has been identified (e.g., Nicholls, 1984; Minnis and Harrison, 1984). Our results showed that the frequency of occurrence of closed MCC exhibits a pronounced daily cycle, with a maximum during night and/or early morning. On the other hand, the diurnal cycle of open MCC is almost absent. The difference between the two morphologies might occur because

open MCC clouds are particularly influenced by large-scale surface forcing, while closed MCC clouds are more affected by longwave cloud-top cooling outside the subtropics (Kazil et al., 2014; Wood, 2012). In this sense, the diurnal cycle of closed MCC clouds is strongly influence by the incoming solar radiation. At night, in the absence of solar forcing, the MABL can become well mixed, and the cloud deck commonly thickens with the renewed access to moisture from the ocean surface (e.g., Lang et al., 2020; Nicholls, 1984; Minnis and Harrison, 1984). This diurnal cycle and its seasonality are consistent with a

diurnal cycle of precipitation observed over the oceans between 35°S and 50°S (e.g., Dai, 2001; Dai et al., 2007), and at Macquarie Island station (54.62°S, 158.85°E; Lang et al., 2018), where precipitation is significantly more frequent at night and during summer. Previous studies suggest that precipitation arising from MABL is probably making a greater contribution than previously thought (Lang et al., 2018, 2020), so understanding its daily cycle is fundamental to understanding the source of uncertainties in the water budget over the SO (Behrangi et al., 2012, 2014).

Investigation of the distribution of MCC clouds around cyclones and cold fronts showed that in the cold sector of extra-tropical cyclones, closed MCCs move along with the mean flow following open MCC clouds. This is consistent with results found in McCoy et al. (2017) for composites of open and closed MCC around marine cold air outbreaks in the Southern Hemisphere. They found that for the cloud evolution along sea level pressure contours, the closed MCC has the highest frequency at the start of the flow, while open MCC clouds are most frequent to the east.



It appears that the relationship between MCC clouds and SST gradients is stronger than previously reported. While McCoy et al. (2017) shows that the relationship of the extreme temperature contrast between the cold polar air and warmer water favors the development of MABL, our results show that the gradients themselves delimit the distribution of open and closed MCC over the SO. This suggests that closed MCC cloud is possibly more influenced by surface forcing than earlier thought over this region.

The current methodology works well overall, yet the distribution over the northeastern sector of New Zealand presents uncertainties in the classification. With a further increase in training samples in the future over this region and the inclusion of more categories such as disorganized MCC, it is expected that our CNN model can be further improved. Future work using this CNN model will focus on the role of large-scale environmental conditions. In particular, we are interested in studying how spatial organization of MCC clouds contributes to the daily cycle of shallow cumulus clouds and precipitation.

*Data availability.* All Himawari-8 data can be accessed using the public website: https://www.eorc.jaxa.jp/ptree/index.html

*Author contributions.* FL prepared the manuscript and performed most of the data analysis. FL and LA implemented the method to train the network model. FL prepared the training data. LA compiled the training dataset. All coauthors provided editorial feedback on the manuscript.

*Competing interests.* The authors declare that they have no conflict of interest.

*Acknowledgements.* This work is supported by Australian Research Council Discovery Projects DP190101362. We thank Daniel Robbins
for help with the artificial neural network.



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





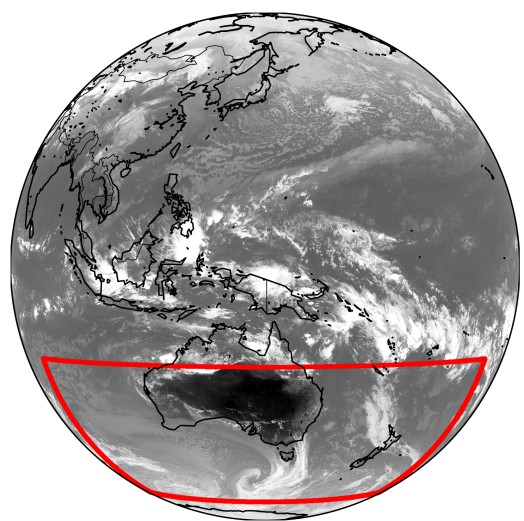

**Figure 1.** A full disk image of Himawari-8 Channel 11 on 15 February 2017 and the domain extent over the Southern Ocean outlined by the red line.

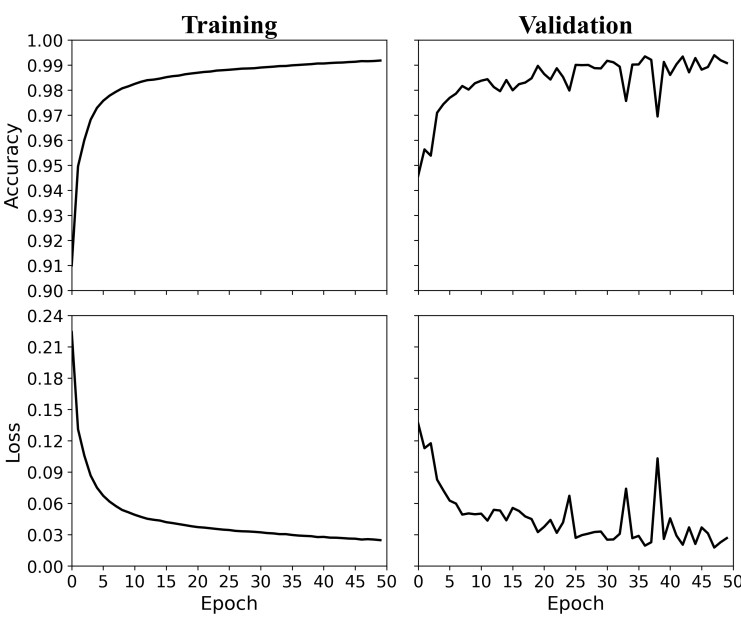

**Figure 2.** Training (left panels) and validation (right panels) accuracy and loss trajectories.





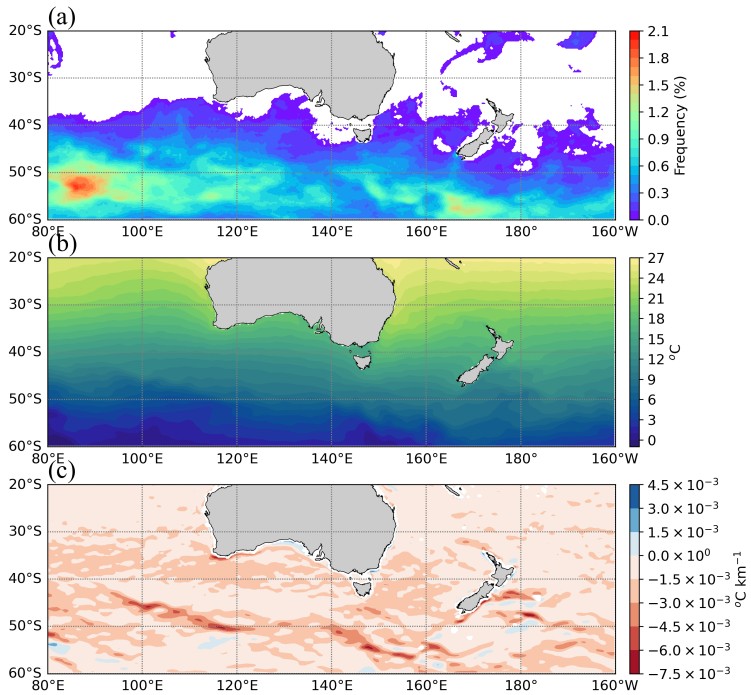

**Figure 3.** (a) Frequency (%) of winds exceeding 20 m s$^{-1}$ from ERA5, (b) mean SST from ERA5 reanalysis and the corresponding (b) SST gradient. Period between 2016 and 2018.

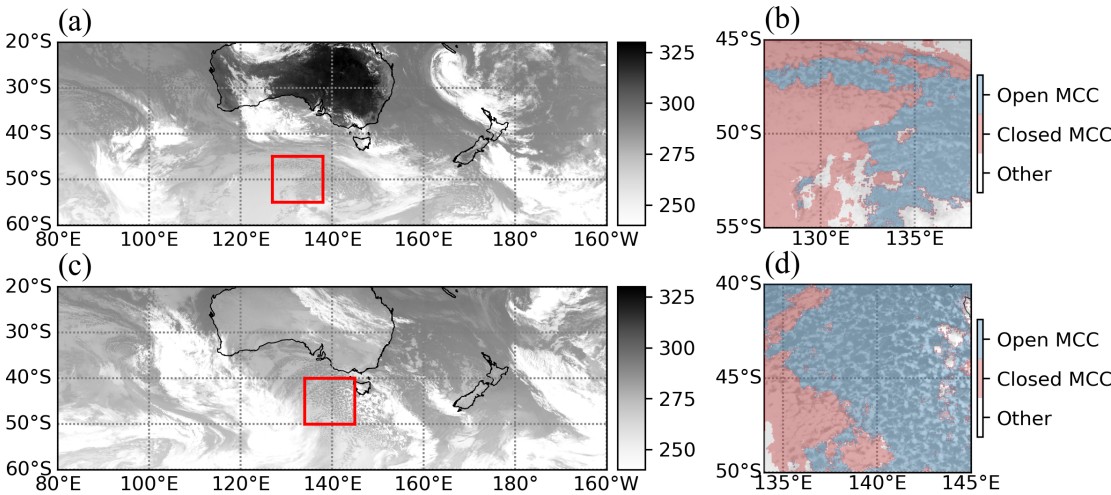

**Figure 4.** Example scenes of AHI Himawari-8 (brightness temperature, channel 11) and MCC clouds identified by the CNN. (a,b) Summertime on 17 Feb 2018 at 02:00 UTC and (c,d) wintertime on 10 Jun 2016 at 21:00 UTC. Red squares delimit the zoom in subscenes.





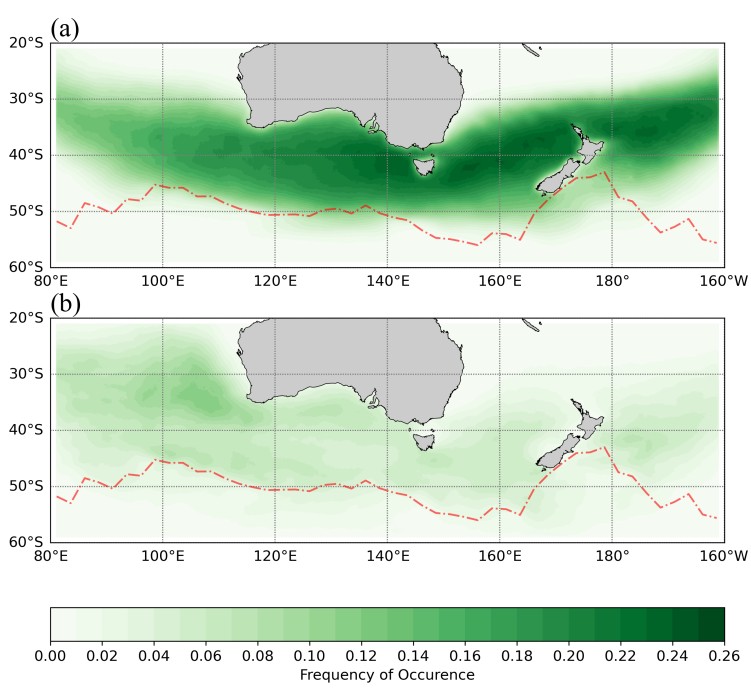

**Figure 5.** Distribution of the frequency of occurrence of MCC cloud type for the period 2016-2018. (a) Open MCC and (b) closed MCC. Red lines indicate the position of the polar front derived from ERA5 SST gradients.



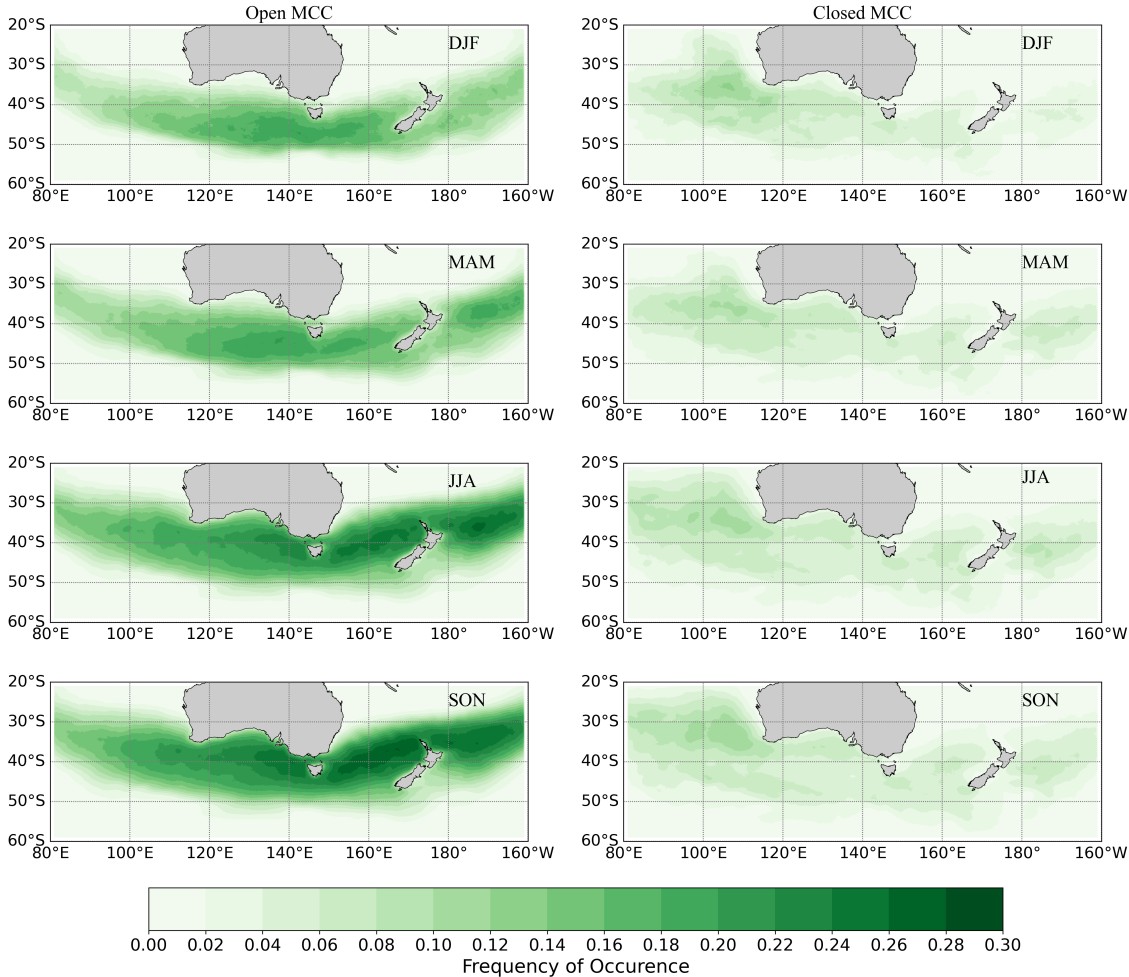

**Figure 6.** Seasonal cycle of the frequency of occurrence of MCC types for the period 2016-2018. Shown are open MCC (left) and closed MCC (right). Seasonal means are shown for winter (DJF), spring (MAM), summer (JJA) and fall (SON).



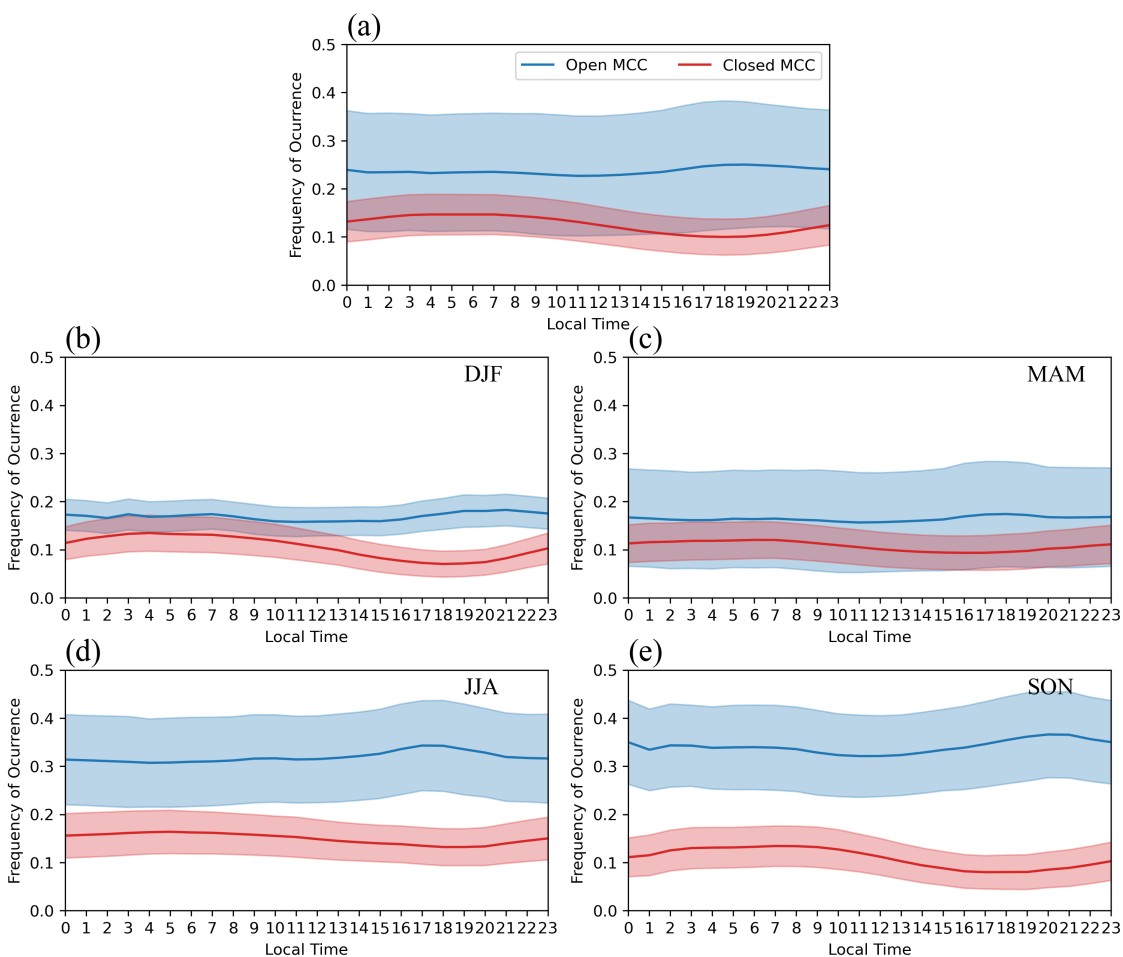

**Figure 7.** Diurnal cycle of the frequency of occurrence of MCC types for the period 2016-2018. Shown are open MCC (blue) and closed MCC (red). Seasonal means are shown for summer (DJF), autumn (MAM), winter (JJA) and spring (SON). Shadings represent one standard deviation. Frequencies are calculated for the latitudinal band between 40° and 50°.





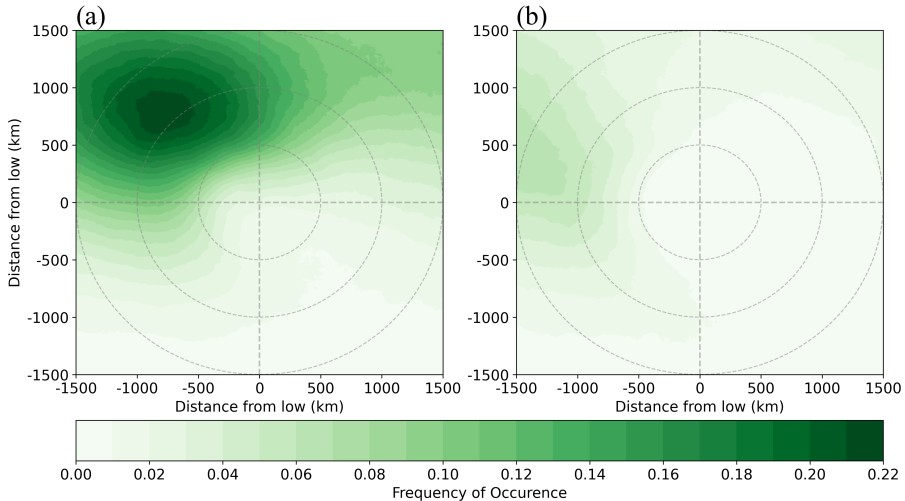

**Figure 8.** Distribution of open and closed MCC in the context of the composite extra-tropical cyclones. Concentric circles indicate distances of 500, 1000, and 1500 km from cyclone center.



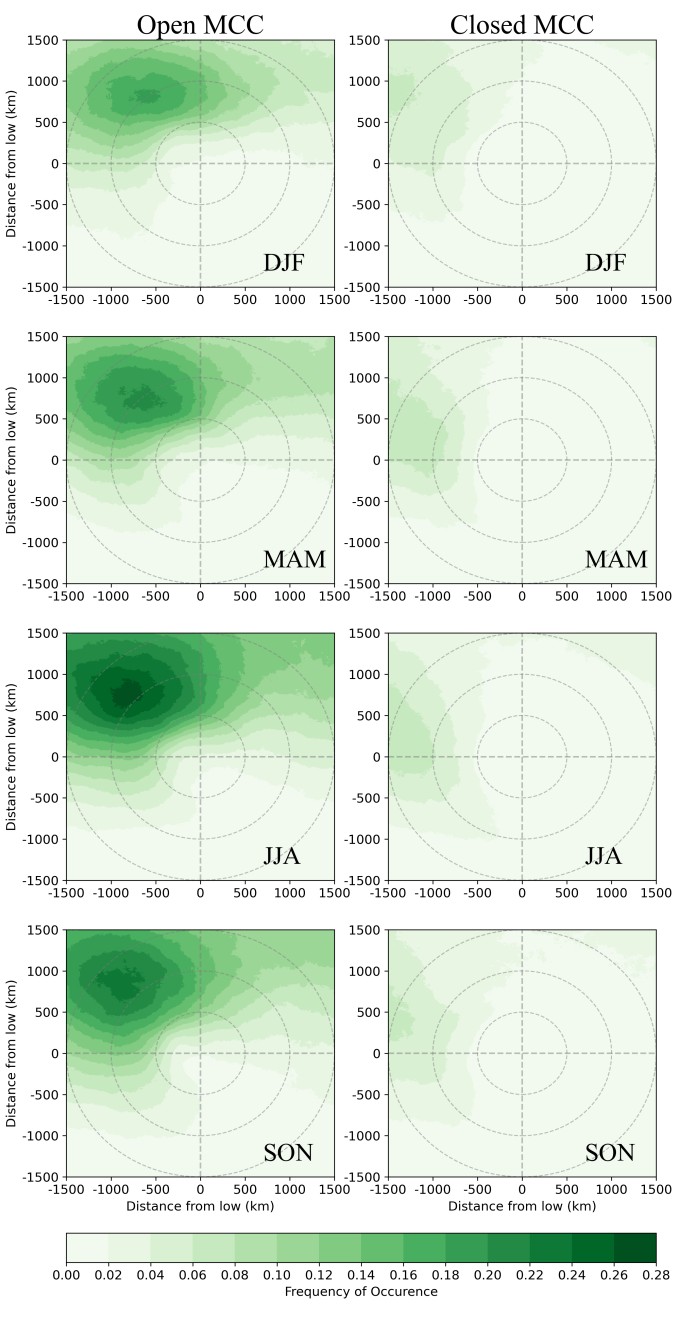

**Figure 9.** Seasonal distribution of open and closed MCC in the context of the composite extra-tropical cyclones. Shown are open MCC (left) and closed MCC (right). Seasonal means are shown for winter (DJF), spring (MAM), summer (JJA) and fall (SON). Concentric circles indicate distances of 500, 1000, and 1500 km from cyclone center.





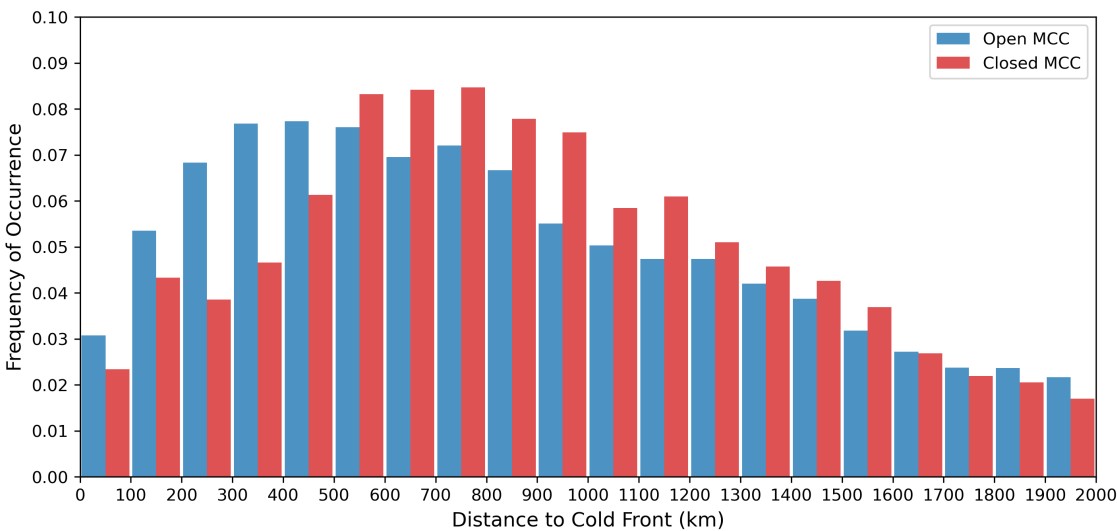

**Figure 10.** Histogram of the relative frequencies of open and closed MCC in the post-cold front sector. Graphs are sorted by distance with 100 km bins.



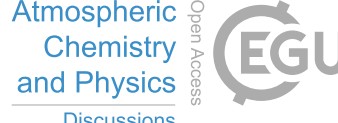

**Table 1.** Confusion matrix of the model predictions on test data

| Categories | Open MCC (Predicted) | Closed MCC (Predicted) | Others (Predicted) |
| --- | --- | --- | --- |
| Open MCC (True) | 0.89 | 0.09 | 0.02 |
| Closed MCC (True) | 0.06 | 0.93 | 0.01 |
| Others (True) | 0.01 | 0.01 | 0.98 |