# Peer review of "A climatology of open and closed mesoscale cellular convection over the Southern Ocean derived from Himawari-8 observations"

_Atmospheric Chemistry and Physics, 2021_

## Referee Comment (RC1)

**Review of "A climatology of open and closed mesoscale cellular convection over the Southern Ocean derived from Himawari-8 observations"**

by Lang et al.

submitted to Atmos. Chem. Phys. Disc.

Summary

The study by Lang et al. presents a first classification and analysis of mesoscale convective organisation in low level clouds on the sub-daily timescale in the Southern Ocean. Utilising geostationary observations by HIMAWARI-8, the authors use a retrieval insensitive to cloud phase unlike previously used retrievals utilised for cloud classifications. The analysis demonstrates the skill of their convolutional neural network approach in identifying open and closed cells. The climatology and spatial relation of open and closed cells to cyclone and cold front activity within this dataset are presented. While some differences to previously published datasets are identified and discussed, the overall preferred occurrence of mesoscale organisation in marine stratocumulus within the cold sector of cyclones and marine cold air outbreaks is confirmed. In addition the importance of the polar front in constraining open- and closed-cell clouds is discussed.

Recommendation

Low-level stratocumulus clouds are the dominant low-cloud type in the midlatitude Southern Ocean region, with many of them being mixed-phase clouds. Many facets of these clouds in this remote region of the world are largely unexplored. At the same time global climate models have been shown to struggle to accurately simulate their cloud-radiative effect. Previous work had emphasised the importance of cloud fraction and mesoscale organisation for cloud field albedo.

The results of this study thus address one of the key uncertainties of low-level clouds in the Southern Ocean and provide new insights regarding their occurrence and the underlying processes driving cloud organisation.

The manuscript is very well written and structured. Furthermore, their findings are discussed in a concise and comprehensive manner. I only have minor queries regarding aspects of their analysis. Once addressed, I can recommend this manuscript to be published in ACP.

General Comment

My main concern in your study is with respect to the training data. As this is a defining aspect of the quality of your CNN, I would like a more comprehensive discussion of:
i) how you generated these data
ii) how you may have introduced an implicit sampling bias by your criteria of identification and scene selection.

P3L82ff: How did you build your training dataset exactly? Why did you choose a combination of all of these variables and how did you implement it?

P4L110: Why did you use brightness temperature as the variable for training the CNN? And why did you train the CNN in channel 11, but identify the training dataset in channel 10?

P4L117: This means that each point is used multiple times in a classification. Once as center point and the other times it is part of the classification for its neighbouring 7 points in each direction. Are these overlaps considered in your overall classification? Or is each point only classified once as center?

P5L128ff:

Were your cloud scenes only identified by only one person? Are their concerns with objectiveness in scene identification (e.g. Stevens et al. 2020)?

Stevens, B, Bony, S, Brogniez, H, et al. Sugar, gravel, fish and flowers: Mesoscale cloud patterns in the trade winds. *Q J R Meteorol Soc*. 2020; 146: 141– 152. https://doi.org/10.1002/qj.3662

Based on your description you identify open-cell as "stringy" clouds. Does this mean that you only picked scenes of low cloud fraction as open cells? Do you observe most cloud fractions in both regimes (as in McCoy et al. 2017) or are they distinctly separated? I.e. you only sample low-cloud fraction open cells and high-cloud fraction closed cells?

How are your 400 independent cloud scenes split across open, closed and "nothing"? And how did you split these scenes into a seperate training and evaluation dataset?

P9L267:

These findings with respect to a diurnal cycle are really interesting and completely novel. It seems consistent with the effects of increased SW insolation, partially compensating cloud-top radiative cooling.

This links back to my question posed on cloud-fraction sampling in your scene identification. It is conceivable that scenes with a higher cloud fraction are more susceptible to this process. Thus, if your identified open-cell clouds are generally characterised by low cloud fraction and little detrained cloud, your results for open-cell stratocumulus may be impacted by this selection?

Specific Comments/Edits

P1L17: "These biases…" Results by Zelinka et al. (2020) suggest that this bias has been "fixed" in many of the new generation climate model runs. While it is not clear how physical the individual approaches of the individual models are, the drastic shortwave bias seems to have been compensated for in some of them. This may be worth mentioning here for completeness.
Reference: Zelinka, M. D., Myers, T. A.,McCoy, D. T., Po-Chedley, S.,Caldwell, P. M., Ceppi, P., et al. (2020).Causes of higher climate sensitivity in CMIP6 models. Geophysical Research Letters, 47, e2019GL085782. https://doi.org/10.1029/2019GL085782.

P2L35: "Overnight…" I am not sure there is a strong diurnal cycle in surface moisture fluxes. Isn't it more the absence of solar heating which partially compensates the LW cooling driving turbulence, that allows these clouds to recover?

P4L109: In line of transparancy it would be helpful if you state which CNN you used (i.e. python package, ect.)

P5L140: The figure suggests that the maximum accuracies for training and validation are at 99%. Please clarify/rephrase.

P5L142: What are the 89% the average of? In your table it says 89% for open and 93% for closed.

P5L143: Following Figure 5, open MCC are far more frequent than closed MCC, so how come they have the lowest training sample size?

P5L148: Are these surface winds? Near-surface winds? Please clarify.

P5L150: I fully agree with your statement. You may present it more convincingly by using a lower wind speed threshold in your figure, or showing a different height of wind speed. Its difficult to speak of "frequent", when the highest frequency is about 2%

P6L156: Doublecheck hyphen. Should it be "warm-water-cool-air contrast"?

Figure 3c/P6L161ff: It may be nice to add a line to mark the location of the polar front for clarity. Reading on it becomes clear that you actually plot the polar front in Fig. 5.I suggest to either link description of polar front location to that figure, or add the location in the SST gradient plot.
P6L168: This is not clear to me. As I understand you also use the SST gradient exceeding a threshold like in Dong et al 2006? Please clarify.

P7L198: How do these numbers relate the the overall number of identified open and closed cell scenes? Otherwise its hard to get the context of what fraction of these regimes were actually associated with cold fronts.

P7L212: It may be helpfull for reference to know how large these images are? I.e. how many 16x16 point segments are in one image? Apologies if I missed it. I am assuming its several, since you identified 25'654 open-cell systems associated with cold fronts alone.

P8L239: Stippling the region of local minimas in both plots, may be a helpful visual aid of your description and make your point clearer.

P9L252: I agree with this conclusion on uncertainty. In my mind this is also evident in Fig. 4b, where clouds in the Southeast corner of the domain are still classified as open-cell.

P9L253: The seasonal cycle was already addressed in previous work (e.g. McCoy et al. 2017). Its worth adding a comment about how consistent your findings are.

P10L303: Please also provide a distance for the closed-cell location in this comparison for clarity.

---

## Author Comment (AC3)

This paper describes aspects of the climatology (spatial patterns, seasonality and diurnal cycles) of open and closed mesoscale cellular convection (MCC) over the Australasian swath of the Southern Ocean extending from 60E-160W. MCC is classified by applying machine learning to labeled open and closed cellular patches of thermal infrared satellite data from the Himawari satellite. The results document interesting differences in the seasonal cycle of open and closed MCC, with a much stronger seasonal variability in open MCC, peaking in late winter/early spring, and little seasonal cycle in closed MCC. Open MCC displays very little diurnal cycle, whereas closed MCC shows a diurnal cycle typical of that for marine stratocumulus. There is interesting latitudinal structure that appears to relate to the location of the polar SST front. Little MCC is found to the south of this front. MCC frequency is also related to the location of cold fronts, open cells typically occurring closer to the cold front, and closed cells more in the warm sector. The results are very interesting and will be of interest to others in the field. The paper is well written, the methods sound, and I recommend publication in ACP. I have a few technical questions and comments that the authors may wish to consider.

**Potential major issue needing address**: The only major potential issue I see is the use of the thermal IR channel only in diagnosing MCC types. I believe that this is fine for diagnosing open MCC, where there is clear contrast across the scene due to breaks in the clouds. For closed cells, there is very weak thermal IR contrast, and so my question is how well one can separate closed MCC from stratus that does not display MCC. The ML approach of Yuan et al. (2020) finds clear contrasts in the locations of stratus vs stratocumulus, and both prevail in midlatitudes. I think the authors need to at least comment on their choice of not including a stratus-type class in their approach. The authors' method seems to agree with previous results (Rampal and Davies 2020), which infrequent MCC poleward of 60S, but these results use visible imagery that has much more discriminating power to separate closed MCC from stratus. I don't know how stratus are removed from the authors' dataset given the use of thermal IR only. Some explanation is required.

R: We thank you for this comment – we agree. We readily acknowledge the limitations of using an infrared channel in the identification of closed MCC. We have added a further discussion on these limitations at lines 147-149 in the revised manuscript.

We note that we do use the visible channel for the initial identification of ideal open and closed MCC clouds, much like the imagery in Yuan et al. (2020). Having identified ideal cloud type on visible images, we then use the matching infrared observation within the CNN. We accept that, potentially, a classification in the visible channel may be ambiguous in an infrared channel. For this very reason, our sample selection criteria is very conservative, only using samples where it was clearly possible to observe closed MCC clouds, as can be seen in the example on new Fig. 2 of the revised manuscript.

The physical consistency of our results, however, suggest the potential of this methodology. Certainly, future refinements may be possible to distinguish beyond clear open and closed MCCs.

We readily accept that different neural network constructions, using different observations, will have their own strengths and weaknesses. One advantage of using the infrared channels is that we can now study the full daily cycle of these clouds, taking benefit of the high-temporal resolution of Himawari-8 (lines 77-79). And using a geostationary platform allows us to consider fixed geographic features such as the relationship with the polar ocean front or the influence of Tasmania or New Zealand.

We also accept that a separation between stratus and stratocumulus would have allowed a more in-depth analysis of the different morphologies of low-level clouds. However, we chose to prioritize closed and open MCC clouds, since on the one hand, this allowed us to simplify the processing and training of the CNN (as indicated at lines 114-115) and, on the other hand, open and closed MCCs have been the primary clouds associated with the dominant synoptic conditions over the Southern Ocean (cold fronts, extra-tropical cyclones and cold-air outbreaks) as shown by McCoy et al. (2017), and as we expressed at lines 44-47.

In our conclusions, we acknowledge the limitations of our neural network in the identification of disorganised MCC and that in future research we hope to improve the sampling process and the inclusion of more categories such as disorganized MCC and no MCC (stratus), to improve our CNN model (lines 417-421). Despite the limitations mentioned above, we believe that our CNN model performs well and is able to separate open and closed MCC clouds..

**Specific comments**

1.  Line 33: Rozendaal et al. (1995) is for me a classic paper on low cloud diurnal cycles and is well worth citing.

R: This is a great suggestion. We have added Rozendaal et al. (1995) as reference in the examples of a diurnal cycle in stratocumulus clouds in the Atlantic and Pacific Oceans.

2.  Line 77. Provide references that the "largest model bias has been linked to this sector"

R: Thank you for this suggestion. We have added a couple of references for this sentence in the revised manuscript.

3.  Line 99: Advanced rather than advance.

R: Suggested revision made.

4.  Section 2.2.1: Some additional information is needed regarding how high clouds and multilevel clouds are screened out using the data. In addition, some basic statistics on the frequency of open and closed cells, high clouds, stratus clouds, multilevel cases, etc would be very helpful.

R: We have expanded the methodology at lines 113-114, commenting specifically on how high and mid-level clouds are classified as "other" in the training data by default, along with stratus, disorganized MCC and all other clouds.

5.  Section 2.2.2: Provide some examples of visible imagery for the open and closed MCC.

R: Thank you for this suggestion. We have added a new figure to show examples of the sample selection for the neural network training (Fig. 2 in the revised manuscript). We have included Channel 11 in this figure because is the main input for the CNN.

6.  Fig 3. Why not show visible imagery rather than thermal IR? I can't tell if the overcast clouds are closed MCC or just stratus.

R: Visible imagery is now added in a new Figure 2. Our intention in Figure 3 (Fig 4. in the revised manuscript) was to show examples of two different seasons, and day and night

conditions. As the analysis of the daily cycle of these clouds is a major part of this study, we consider that showing a thermal IR channel for nighttime is particularly important.

Line 195. A number of degrees is missing in my pdf.

R: Thanks for noticing this error. We have corrected this in the revised manuscript.

7. Line 198. What exactly constitutes a "system"? Is this an 80x80 km patch? Are these systems really independent if you measure them every 15 minutes?

R: A MCC system is defined as a continuous group of grid points classified as either closed or open MCCs, we use the term system to refer to an event at a specific time, rather than a complete life cycle (lines 217-219). We have specified in the revised manuscript that a system refers to an event at a specific time, rather than a complete life cycle. An 80×80 km patch corresponds to the size of the window used as a sample in the CNN training (a window of $16 \times 16$ grid points). Regarding to the independency of the systems, samples 15 minutes apart were not measured, all the training data have been randomly sampled across time, with at least 12 hours separation between samples.

8. The lack of diurnal cycle in open MCC frequency is very interesting and novel. However, what would be even more interesting is whether the diurnal cycle of cloud cover (rather than frequency) in open MCC exhibits a diurnal cycle. Do the authors have cloud mask data that can be used to determine this?

R: We agree that it would be interesting to analyse the daily cycle of the fractional cloud cover within the open MCC. Does the fractional cloud cover of open MCC change over the course of the day?. We are intrigued by this idea, but it is not immediately possible within our current methodology. We would need to estimate how the Himawari-8 view angle affects the fractional cloud cover. Possibly, we could correct for viewing angle using MODIS, but this is outside the scope of this paper. We thank the reviewer for this idea.

**References**

Rozendaal, M. A., Leovy, C. B., & Klein, S. A. (1995). An Observational Study of Diurnal Variations of Marine Stratiform Cloud. Journal of Climate, 8(7), 1795–1809. https://doi.org/10.1175/1520-0442(1995)008<1795:AOSODV>2.0.CO;2

---

## Author Comment (AC4)

*Summary*

The study by Lang et al. presents a first classification and analysis of mesoscale convective organisation in low level clouds on the sub-daily timescale in the Southern Ocean. Utilising geostationary observations by HIMAWARI-8, the authors use a retrieval insensitive to cloud phase unlike previously used retrievals utilised for cloud classifications. The analysis demonstrates the skill of their convolutional neural network approach in identifying open and closed cells. The climatology and spatial relation of open and closed cells to cyclone and cold front activity within this dataset are presented. While some differences to previously published datasets are identified and discussed, the overall preferred occurrence of mesoscale organisation in marine stratocumulus within the cold sector of cyclones and marine cold air outbreaks is confirmed. In addition the importance of the polar front in constraining open- and closed-cell clouds is discussed.

*Recommendation*

Low-level stratocumulus clouds are the dominant low-cloud type in the midlatitude Southern Ocean region, with many of them being mixed-phase clouds. Many facets of these clouds in this remote region of the world are largely unexplored. At the same time global climate models have been shown to struggle to accurately simulate their cloud-radiative effect. Previous work had emphasised the importance of cloud fraction and mesoscale organisation for cloud field albedo. The results of this study thus address one of the key uncertainties of low-level clouds in the Southern Ocean and provide new insights regarding their occurrence and the underlying processes driving cloud organisation.

The manuscript is very well written and structured. Furthermore, their findings are discussed in a concise and comprehensive manner. I only have minor queries regarding aspects of their analysis. Once addressed, I can recommend this manuscript to be published in ACP.

*General Comment*

My main concern in your study is with respect to the training data. As this is a defining aspect of the quality of your CNN, I would like a more comprehensive discussion of:
    i) how you generated these data
    ii) how you may have introduced an implicit sampling bias by your criteria of identification and scene selection.

P3L82ff: How did you build your training dataset exactly? Why did you choose a combination of all of these variables and how did you implement it?

R: Multiple channels and cloud products from Himawari-8 were analyzed with the goal to intensify scenes with areas where open and closed MCC clouds were clearly present to build the training dataset as objectively as possible. About 400 different scenes with clear open or closed MCC clouds were selected, taking care to have about the same amount of samples per season and during daytime and nighttime. Then the areas of definite open MCC, closed MCC, and others were "cropped" by hand from these scenes, and the latitude, longitude, and time of each pixel contained inside these areas were recorded per category. This constituent the raw training dataset, consisting of four fields (latitude, longitude, time, and category) and ~2.7 million records (pixels). The dataset used to train the CNN was created using the raw training dataset and the desired size of the moving window. The CNN was trained using different infrared channels from Himawari-8. We found that Channel 11 showed the highest accuracy of all infrared channels and combinations we tried. Adding the solar and satellite angles for the center pixel showed significant improvement of model

accuracy at a very minor computational cost; we hypothesize that the improvement is due to the insolation effects and deformations related to the orthogonal projection, which are mitigated by providing the model this information during training. The final training dataset consisted of normalized two dimensional (16 x 16 windows) arrays of Himawari-8 Channel 11, and four normalized scalars (two solar angles and two satellite angles). The dataset was separated into training and validation datasets with a ratio of 80:20, respectively. Once the CNN model had been trained, we predicted the three categories in each hourly Himawari-8 image between 2016 and 2018.

We have rewritten and expanded Section 2 for a better explanation of the sampling process and CNN training.

P4L110: Why did you use brightness temperature as the variable for training the CNN? And why did you train the CNN in channel 11, but identify the training dataset in channel 10?

R: We used several products and channels from Himawari-8 as contextual information to identify open and closed MCC clouds (including channel 10, other infrared channels and visible), but the main input for the CNN is Channel 11. We use specifically Channel 11, first because it is an infrared channel, which allows us to take advantage of the high-temporal resolution of Himawari-8; and second, as we stated in lines 124-125, we tested a variety different infrared channels (and combinations) as inputs, with Channel 11 having the best performance.

For a better explanation of the sampling process and CNN training, we have rewritten Section 2 in the revised manuscript.

P4L117: This means that each point is used multiple times in a classification. Once as center point and the other times it is part of the classification for its neighbouring 7 points in each direction. Are these overlaps considered in your overall classification? Or is each point only classified once as center?

R: This point is correct, each point is classified using the information of itself and the surrounding pixels. The overlaps are not considered, each point is only classified as once as a center.

P5L128ff:
Were your cloud scenes only identified by only one person? Are their concerns with objectiveness in scene identification (e.g. Stevens et al. 2020)?

Stevens, B, Bony, S, Brogniez, H, et al. Sugar, gravel, fish and flowers: Mesoscale cloud patterns in the trade winds. Q J R Meteorol Soc. 2020; 146: 141– 152. https://doi.org/10.1002/qj.3662

R: The sampling selection process and labeling of the three categories were carried out by a team of three co-authors, but discussed widely through our research group. We understand and respect the reviewer's concern about the objectiveness; however, as it is a manual process (supervised training), it is implicit that there are uncertainties/biases associated with expert criteria in the labelling. Compared to the Max Planck Institute for Meteorology's study (Stevens et al. 2020), we simply do not have the same resources or the same number of trained scientists for the sampling/training process. For this very reason, our sample selection process is very conservative, only using samples where it was clearly possible to observe

open and closed MCC clouds, as can be seen in the examples on new Fig. 2 of the revised manuscript.

Based on your description you identify open-cell as "stringy" clouds. Does this mean that you only picked scenes of low cloud fraction as open cells? Do you observe most cloud fractions in both regimes (as in McCoy et al. 2017) or are they distinctly separated? I.e. you only sample low-cloud fraction open cells and high-cloud fraction closed cells?

R: We have rewritten the sentence where we mentioned the criteria to identify open MCCs. We do not select scenes only with low-cloud fraction as open MCC or high-cloud fraction as closed MCC clouds. We selected open and closed MCCs strictly on visible structure/geometry, independent of fractional cloud fraction. Open MCCs were identified as ring of low-level clouds arranged in rings with a clear region in the center. While closed MCCs were identified by low-level clouds organized into distinctive patterns of shaped cells with clear edges. We have added a new figure (Fig. 2 in the revised manuscript) to show an example of both categories. In addition, we have rewritten the Section 2 to indicate that different Himawari-8 channels and cloud products are used to identify closed and open MCC clouds.

How are your 400 independent cloud scenes split across open, closed and "nothing"? And how did you split these scenes into a seperate training and evaluation dataset?

R: We have specified that the ~400 independent scenes are selected for each category (lines 151-152). We have also added the percentages used for training and evaluations of this dataset (lines 157-158).

P9L267:

These findings with respect to a diurnal cycle are really interesting and completely novel. It seems consistent with the effects of increased SW insolation, partially compensating cloud-top radiative cooling.

This links back to my question posed on cloud-fraction sampling in your scene identification. It is conceivable that scenes with a higher cloud fraction are more susceptible to this process. Thus, if your identified open-cell clouds are generally characterised by low cloud fraction and little detrained cloud, your results for open-cell stratocumulus may be impacted by this selection?

R: This is a very intriguing question and somewhat similar to that of Reviewer 1 regarding the potential for a diurnal cycle of the fractional cloud cover for open MCC. Open MCC do, in general, have a lower fractional cloud cover than closed MCC (Rampal and Davis, 2020; McCoy et al. 2017), and this may potentially help explain the weaker diurnal cycle in the open MCC when compared to the closed MCC. But as clarified, we selected open and closed MCCs strictly on visible structure/geometry, independent of fractional cloud fraction. Ultimately, we can not answer this question in any quantitative sense with this methodology.

Similar to the response to reviewer 1, the varying view angle coupled with variable cloud-top height complicates the calculation of the local fractional cloud cover for both open and closed MCC. An immediate way to explore this would be to study the life cycle of cold air outbreak events with Himawari 8 imagery and the CNN.

*Specific Comments/Edits*

P1L17: "These biases…" Results by Zelinka et al. (2020) suggest that this bias has been "fixed" in many of the new generation climate model runs. While it is not clear how physical the individual approaches of the individual models are, the drastic shortwave bias seems to have been compensated for in some of them. This may be worth mentioning here for completeness.

Reference: Zelinka, M. D., Myers, T. A.,McCoy, D. T., Po-Chedley, S.,Caldwell, P. M., Ceppi, P., et al. (2020). Causes of higher climate sensitivity in CMIP6 models. Geophysical Research Letters, 47, e2019GL085782. https://doi.org/10.1029/2019GL085782.

R: Thank you for this suggestion. We have added Zelinka et al. (2020) as reference in the Introduction.

P2L35: "Overnight…" I am not sure there is a strong diurnal cycle in surface moisture fluxes. Isn't it more the absence of solar heating which partially compensates the LW cooling driving turbulence, that allows these clouds to recover?

R: We certainly did not mean to imply that there is a diurnal cycle of surface moisture fluxes. We do not have measurements to confirm this. We tried to express is that at night, in the absence of solar forcing, the boundary layer can once again become well mixed, and the cloud deck commonly thickens with the renewed access to moisture from the ocean surface. We have rewritten this sentence and added a reference for completeness (lines 37-38).

P4L109: In line of transparancy it would be helpful if you state which CNN you used (i.e. python package, ect.)

R: Thank you for this suggestion. We have indicated that the CNN model was built using the specific python package called TensorFlow (line 121).

P5L140: The figure suggests that the maximum accuracies for training and validation are at 99%. Please clarify/rephrase.

R: Thanks for pointing out this mistake. The original figure corresponded to a training that produced unrealistic high accuracies due to an error on the implementation of the training dataset. It was included in the manuscript by accident as the shape of the correct figure is almost the same. The figure has been replaced to show the correct data (Fig. 3 in the revised manuscript), which is consistent with the confusion matrix (Table 1), as you rightly pointed it should be. The accuracy and loss series shown are the average over all classes (open MCC, closed MCC, and others) per each training epoch.

P5L142: What are the 89% the average of? In your table it says 89% for open and 93% for closed.

R: The percentages correspond to the accuracy series averaged for each individual category. We have rewritten the sentence to clarify the meaning of the percentages.

P5L143: Following Figure 5, open MCC are far more frequent than closed MCC, so how come they have the lowest training sample size?

R: We have rewritten this sentence to indicate that ~1.2 million individual pixels are under the open MCC category, ~0.6 million under the closed MCC category, and the remaining (~0.9 million) under the others category (lines 156-157). However, the frequencies show in Figure 5 are independent of the sampling size of each categories. This is because the differences in frequencies between open and closed MCC are more related to whether the samples are sufficiently representative of each cloud type, so the CNN can identify them rather than the number of pixels for each category. Our objective was to take into account a wide range of synoptic meteorology, solar zenith angles, seasonal and diurnal variation, independent of whether one category had more labeled pixels than another.

P5L148: Are these surface winds? Near-surface winds? Please clarify.

R: Effectively, the winds correspond to near-surface wind from ERA5. We have added this to the revised manuscript (line 169 and Fig. 4).

P5L150: I fully agree with your statement. You may present it more convincingly by using a lower wind speed threshold in your figure, or showing a different height of wind speed. Its difficult to speak of "frequent", when the highest frequency is about 2%

R: Thank you for this suggestion. We have changed the wind speed threshold. The Fig. 4 in the revised manuscript shows the wind speed frequencies in the range of 0-25 %.

P6L156: Doublecheck hyphen. Should it be "warm-water-cool-air contrast"?

R: Suggested revision made.

Figure 3c/P6L161ff: It may be nice to add a line to mark the location of the polar front for clarity. Reading on it becomes clear that you actually plot the polar front in Fig. 5.I suggest to either link description of polar front location to that figure, or add the location in the SST gradient plot.

R: Suggested revision made. We have added the polar front to Fig. 4 in the revised manuscript.

P6L168: This is not clear to me. As I understand you also use the SST gradient exceeding a threshold like in Dong et al 2006? Please clarify.

R: We have added a sentence to explain that our definition of polar front does not consider a minimum threshold. We define the polar front simply as the southernmost maximum in the SST gradient (lines 186-187).

P7L198: How do these numbers relate the the overall number of identified open and closed cell scenes? Otherwise its hard to get the context of what fraction of these regimes were actually associated with cold fronts.

R: For the period of time analyzed, we used approximately 26,280 satellite images. From this, approximately $1.5 \times 10^{10}$ grid points are classified as either open or closed MCC. To reduce the computational cost of the estimation of the open and closed MCC clouds associated with cold fronts, we chose to define systems of open and closed MCCs (lines 216-219) rather than single grid points, this is because overall a cold front is associated with a very large number of grid points. This does not alter our results, as the relationship to the cold fronts is calculated on a point-by-point basis. We have rewritten this sentence to

indicate the number of satellite images used, which gives an idea of the fraction of these regimes associated with fronts (lines 221-222).

P7L212: It may be helpfull for reference to know how large these images are? I.e. how many 16x16 point segments are in one image? Apologies if I missed it. I am assuming its several, since you identified 25'654 open-cell systems associated with cold fronts alone.

R: Good idea. We have added the size of the domain at line 124, which is 801 × 2401 grid points. Therefore, in the complete domain there is space for approximately 7500 windows of 16×16 grid points.

With respect to the case of the open MCC systems, the size of the systems is independent of the size of the sampling window. Furthermore, the number of open MCC systems associated with cold fronts is for the whole study period, which considers hourly data between 2016 and 2018, totaling approximately 26,280 satellite images.

P8L239: Stippling the region of local minimas in both plots, may be a helpful visual aid of your description and make your point clearer.

R: Thank you for this suggestion. We have added a contour line to delimit the local minimas for both open and closed MCCs (Fig. 6 in the revised manuscript).

P9L252: I agree with this conclusion on uncertainty. In my mind this is also evident in Fig. 4b, where clouds in the Southeast corner of the domain are still classified as open-cell.

R: Thank you for this observation. We have expanded the discussion to include this suggestion (lines 276-277)

P9L253: The seasonal cycle was already addressed in previous work (e.g. McCoy et al. 2017). Its worth adding a comment about how consistent your findings are.

R: Thank you for this suggestion. We have added a comparison to McCoy et al. (2017) seasonal cycle in the revised manuscript (lines 283-284)

P10L303: Please also provide a distance for the closed-cell location in this comparison for clarity.

R: Thank you for this suggestion. The value in brackets corresponded to the distance from the closed MCC maximum. In the revised manuscript, we have rewritten this sentence to clarify the range of distances corresponding to the closed MCC maximum (line 330), similarly to the range of distances to the open MCC maximum at line 316.